# Emerging Anticancer Potentials of Selenium on Osteosarcoma

**DOI:** 10.3390/ijms20215318

**Published:** 2019-10-25

**Authors:** Kok-Lun Pang, Kok-Yong Chin

**Affiliations:** Department of Pharmacology, Faculty of Medicine, Universiti Kebangsaan Malaysia, Cheras, Kuala Lumpur 56000, Malaysia; pangkoklun@gmail.com

**Keywords:** bone, cancer, chemoprevention, osteoblasts, oxidative stress, trace elements

## Abstract

Selenium is a trace element essential to humans and forms complexes with proteins, which exert physiological functions in the body. In vitro studies suggested that selenium possesses anticancer effects and may be effective against osteosarcoma. This review aims to summarise current evidence on the anticancer activity of inorganic and organic selenium on osteosarcoma. Cellular studies revealed that inorganic and organic selenium shows cytotoxicity, anti-proliferative and pro-apoptotic effects on various osteosarcoma cell lines. These actions may be mediated by oxidative stress induced by selenium compounds, leading to the activation of p53, proapoptotic proteins and caspases. Inorganic selenium is selective towards cancer cells, but can cause non-selective cell death at a high dose. This condition challenges the controlled release of selenium from biomaterials. Selenium treatment in animals inoculated with osteosarcoma reduced the tumour size, but did not eliminate the incidence of osteosarcoma. Only one study investigated the relationship between selenium and osteosarcoma in humans, but the results were inconclusive. In summary, although selenium may exert anticancer properties on osteosarcoma in experimental model systems, its effects in humans require further investigation.

## 1. Introduction

Selenium is an essential trace element in humans and is a non-metal from Group 16 of the periodic table that shares some similar physicochemical properties with sulphur. This element is rarely found in its elementary state but rather in its inorganic or organic forms in natural compounds [1,2]. The examples of inorganic selenium are selenium dioxide (SeO_2_), selenite (SeO_3_^2−^) and selenate (SeO_4_^2−^), whereas those for organic selenium are selenide, diselenides, selenol or selenothiol and seleninic acid (selenium-based acid; RSeOH) [2,3,4,5].

Selenium can be obtained mainly from food, such as Brazil nuts, garlic, onion, mushroom, broccoli, meat, egg, seafood and internal organs and in negligible quantity from drinking water [6,7,8]. The main selenium species present in food are selenomethionine (SeMet), selenocysteine (SeC), selenium-methylselenocysteine (Se-MSC), SeO_3_^2^ and SeO_4_^2−^ [9]. The organic forms of selenium are present in greater amount in food, and their bioavailability is higher because they are absorbed easily compared with inorganic selenium [7,8,10]. The US-recommended dietary allowance and UK-recommended reference nutrient intake of selenium for an adult are 55 and 60 µg/day, respectively [11,12]. The average dietary intake of selenium varies across countries [10,13,14,15,16]. People from several countries, such as the UK, China, New Zealand and Finland, are traditionally deficient in selenium dietary intake [7,8,10,15,16].

Selenium is vital in various physiological processes and can incorporate into a protein as selenoproteins (SeC-containing protein), selenium-binding proteins (without SeC) and certain proteins rich in cysteine or methionine (such as SeMet), where sulphur is replaced by selenium in a nonspecific manner [1,17]. This element is required as a cofactor in more than 25 selenoproteins including glutathione peroxidase, thioredoxin reductase and iodothyronine deiodinase [1,18]. Selenoproteins are essential in oxidative defence, detoxification, immune response and thyroid hormone regulation [8,18,19]. Minor selenoproteins, such as selenoprotein-P, -W and -R, have antioxidant functions [8,20,21]. Selenoprotein-S is involved in inflammatory response and protein quality control [1,8]. Inadequate selenium uptake is related to cardiovascular diseases (Keshan disease), muscular diseases and bone diseases (osteopenia and Kashin-Beck disease) [2,18,22].

Selenium also exhibits potential in vivo and in vitro anticancer activities. Inorganic and organic selenium induce cell cycle arrest, cytotoxicity and apoptosis induction on cancerous cell lines from colon, breast, lung or prostate origins [2,5,23,24,25,26,27]. High levels of inorganic and organic selenium induce oxidative stress in cancer cells by generating reactive oxygen species (ROS) [5,28]. Selenium also prompts oxidative damage on DNA and mitochondria, leading to mitochondrial dysfunction in caspase-dependent or -independent apoptosis [2]. Organic selenium, such as methyseleninic acid (MSeA) inhibits angiogenesis by downregulating integrin β3 signalling [29]. Inorganic and organic selenium activate calcium influx, endoplasmic reticulum stress, 5’ adenosine monophosphate-activated protein kinase (AMPK), AKT, tumour suppressor protein phosphatase and tensin homolog (pTEN) and p53 signalling upstream of apoptosis events [24,27,30,31,32]. The potency of selenium in cytotoxicity depends greatly on chemical forms, cancer cell types and bioavailability [5]. With the use of human findings, the Nutritional Prevention of Cancer (NPC) trial showed that selenium intake decreases the risk of lung, colorectal and prostate cancer [33,34,35]. The Linxian Nutritional Intervention Trials also reported that a high level of basal serum selenium is associated with a marked reduction in mortality among patients with oesophageal and gastric cancer [33,36].

Bone cancer is one of the most prevalent cancers with the lowest long-term survival rate among children and adolescents [37]. Osteosarcoma is the most common malignant primary bone tumour in children and teenagers with 400 new cases annually in the US [38]. Its annual incidence rate is relatively low, with approximately 2–3.4 per million people worldwide [39,40], as compared with the most common childhood malignancy, childhood leukaemia (46.7 per million people) [41]. Osteosarcoma can be classified histopathologically into low-, intermediate- and high-grade, and more than 90% of osteosarcomas are high-grade malignancies [42]. Osteosarcoma is commonly found at the metaphyseal of long bones and its aetiology remains unknown [43]. Patients with non-metastatic osteosarcoma have a 65–70% five-year survival rate, whereas those with metastatic osteosarcoma have a poor survival rate of 19–30% [38,42]. The relapse rate for osteosarcoma is high, whereby nearly 40% of patients initially diagnosed with non-metastatic disease develop recurring osteosarcoma [44]. The current treatment options for osteosarcoma include surgery, radiotherapy and chemotherapy [38,40,45]. Adjuvant chemotherapy can greatly improve the quality of life and 5-year survival rate of these patients, but 30–50% of them still die because of lung metastasis [43,46]. Selenium is crucial in bone metabolism and health [47,48]. The therapeutic effect of selenium against osteosarcoma has been widely explored. Given its anticancer potential, selenium may serve as a prophylaxis or therapeutic option for osteosarcoma. Therefore, this review aims to summarise the anticancer effects of selenium in primary osteosarcoma from pre-clinical and clinical studies.

## 2. Literature Search

The literature search was performed between 1^st^–25^th^ March 2019 on PubMed and Scopus using the keywords ‘Selenium’ OR ‘Selenite’ OR ‘Selenate’ OR ‘Selenide’ OR ‘Selenol’ OR ‘Organoselenium’ OR ‘Selenocysteine’ OR ‘Selenomethionine’ OR ‘Selenoprotein’ AND ‘Osteosarcoma’. We also examined the reference lists of the retrieved articles. Original research articles regarding the anticancer effects of selenium in osteosarcoma published in English were included. A total of 20 relevant studies were included in this review.

## 3. Evidence from In Vitro Study

### 3.1. Selenium Exerts Cytotoxicity, Anti-Proliferation and Pro-Apoptotic Activities on Osteosarcoma Cells

Inorganic and organic forms of selenium including SeO_2_ [49], SeO_3_^2−^ [50], selenium-polysaccharide (Se-Poly) [51], MSeA [52,53], Se-MSC [54] and SeC [55] induce cytotoxicity, anti-proliferation and apoptosis on osteosarcoma cells. These anticancer properties of inorganic and organic selenium were studied using mouse osteosarcoma K7M2 cells [56,57] and several human osteosarcoma cell lines, such as MG-63 [49,54,55,58,59,60], U-2 OS [49,50,51,52,53,61], Saos-2 [49,62], drug-resistant Saos-2/MTX300 [54], 143B [63,64,65] and MNNG/HOS cells [66]. SeO_3_^2−^ concentration (as low as 10 µM) inhibits the 12−day colony formation activities of U-2 OS cells [50]. Organic selenium, such as SeC and Se-MSC, also induces cell cycle arrest and sub-G1 accumulation in MG-63 cells [54,55]. SeC-induced S-phase arrest is associated with the downregulation of cyclinA and cyclin-dependent kinase-2 (CDK-2) proteins [55]. Most of the cytotoxicity and anti-proliferation effects of selenium species are time-, concentration- or cell type-dependent [49,50,51,54,55]. For example, MSeA significantly increases the level of necrosis in U-2 OS cells within a longer treatment time period (48 h) [52]. Se-MSC inhibits the growth of MG-63 cells, but enhances that of U-2 OS cells in a concentration-independent manner [54]. This finding indicates that each selenium species may exert different effects on osteosarcoma cell models. At present, the most effective selenium species against osteosarcoma is difficult to determine on the basis of cellular findings. Direct comparison is impossible due to differences in treatment time, concentration and osteosarcoma cell models used. To date, only two studies reported the IC_50_ value of selenium (SeO_3_^2−^ and MSeA) in osteosarcoma cells [50,52]. Other selenium species such as SeO_4_^2−^, selenide, selenol and selenoproteins have not been tested on osteosarcoma.

### 3.2. Molecular Mechanism of Selenium-Induced Osteosarcoma Cell Death

Various forms of selenium species, including SeO_2_ [49], SeO_3_^2−^ [50], Se-MSC [54] and SeC [55] induce apoptosis of human osteosarcoma cells by showing typical apoptotic morphological and ultrastructural changes. However, the molecular mechanism of selenium-induced cell death is poorly characterised. Selenium-induced osteosarcoma cell death may be modulated via ROS production. SeO_3_^2−^ [50] and SeO_3_^2−^-substituted hydroxyapatite (HA) nanoparticles (SeHAN) [60,66] induce early ROS production in U-2 OS, MG-63 and MNNG/HOS cells. SeC also induces ROS production as shown by dichlorodihydrofluorescein diacetate and Mito-SOX staining [55]. ROS is an important modulator in p53 activation [67,68]. SeC also induces early p53 activation by phosphorylating the Ser 15, Ser 20 and Ser 392 sites of p53 protein [55]. Furthermore, ROS upregulation is crucial because glutathione pretreatment almost completely abrogates SeC-induced apoptosis and partially reduces p53 phosphorylation [55]. The crosstalks among p53, ataxia-telangiectasia mutated kinase (ATM) and forkhead box O3a (FOXO3a) have been identified [69]. However, MSeA induces U-2 OS cell death via ATM (Ser1981) and FOXO3a activation that is independent of ROS induction and phosphorylated H2A histone family member X (γH2AX) activation [52,53]. MSeA, not SeO_3_^2−^, induces the nuclear translocation of FOXO3a protein [53]. The knockdown of ATM (by KU55933 inhibitor) and FOXO3a (by hairpin RNA vector transfection) further suppressed MSeA-induced cytotoxicity [53]. Additionally, Werner Syndrome protein (WRN) serves as a potential combinational target, and its downregulation further increased the MSeA-induced U-2 OS cell death [52]. A comprehensive molecular study is required to elucidate the upstream molecular mechanisms of different selenium species in p53/ATM/FOXO3a/WRN signalling pathway axis.

SeC also induces early mitochondrial dysfunction via mitochondrial fragmentation (from protonema to punctiform phenotype) and mitochondrial membrane potential loss, a possible result of p53 activation [55]. Additionally, proapoptotic proteins Bax, Bad and pTEN are transcriptionally upregulated by p53 activation [70,71]. In line with this finding, SeO_3_^2−^ [50], Se-MSC [54] and SeC [55] also downregulate the antiapoptotic Bcl-2 and Bcl-XL protein and upregulate the Bax and Bad protein in osteosarcoma cells. SeO_3_^2−^ treatment significantly increases the *P53* and *PTEN* mRNA levels [50]. Caspase-9 is activated upon the assembly of the apoptosome after increase in the level of Bax or Bad protein, mitochondrial dysfunction and the release of mitochondrial apoptotic proteins, such as cytochrome *c* [70]. The activated caspase-9 subsequently activates the downstream executioner caspases (caspase-3, -6 and -7) and cleaves cellular proteins including poly(ADP-ribose) polymerase (PARP) [72,73]. SeC activates caspase-9 and caspase-3/7 and induces PARP cleavage [55]. SeO_3_^2−^ also increases *CASP9* and *CASP3* mRNA levels and caspase-3 protein level [50]. The mRNA levels for *CASP6* and *CASP8* are unaffected by SeO_3_^2−^ treatment [50].

The current understanding of the molecular mechanisms of selenium-induced osteosarcoma apoptosis is summarised in Figure 1.

### 3.3. Role of Selenium in the Tumour Microenvironment

The bone tissue is made up of several cells, including osteoblasts, osteoclasts, chondrocytes, mesenchymal stem cells, blood cells and endothelial cells. The osteoclasts play an important role in the pathogenesis of osteosarcoma, especially in the progression and metastasis of the cancer [74]. Osteoclasts formation and activation are tightly regulated through both receptor-activation of nuclear factor κB (RANK) ligand (RANKL) and macrophage-colony stimulating factor [75]. The naturally-occurring inhibitor of RANKL, osteoprotegerin (OPG) synthesised by osteoblasts, protects against bone loss by acting as a negative regulator for RANK/RANKL pathway [75]. The RANKL is secreted by osteocytes, osteoblasts and mesenchymal stem cells, as well as osteosarcoma cells [76,77]. RANKL activates its receptor, RANK, and then triggers the bone resorption by osteoclasts, which in turn promotes invasion and metastasis of osteosarcoma [74]. RANKL inhibitors such as denosumab (currently in phase II clinical trial) may be effective against osteosarcoma [78,79]. Thus far, the effects of selenium on the RANK/RANKL/OPG pathway remain inconclusive. SeO_3_^2−^ suppresses RANKL-induced osteoclastogenesis through inhibition of ROS-induced signalling pathways in mouse bone marrow-derived monocytes and RAW 264.7 cell line [48,80]. Additionally, selenium (unknown species) was reported to inhibit the transcription activity of RANKL and downregulate the mRNA levels of OPG in human osteosarcoma Saos-2 cells [81] More research is required to confirm the effects of selenium in the RANK/RANKL/OPG pathway.

Additionally, mesenchymal stem cells serve as an important modulator in the pathogenesis of osteosarcoma, wherein it supports the osteosarcoma progression and metastasis via the secretion of cytokines/growth factors [74,82]. Mesenchymal stem cells can be targeted in osteosarcoma treatment via the restoration of cytokines/growth factors signalling, or transforming the stem cells into mature osteoblasts [82]. The effects of selenium on mesenchymal stem cells are not conclusive. SeO_3_^2−^ was reported to protect bone marrow stromal cells against hydrogen peroxide-induced inhibition of osteoblastic differentiation through inhibiting oxidative stress and ERK activation [83]. A ruthenium (II) functional selenium nanoparticles and citrate functionalised selenium nanoparticles were reported to promote the proliferation and osteogenic differentiation of human umbilical cord mesenchymal stem cells [84]. Recently, Ahmed et al. demonstrated that 48 h of SeO_2_ nanoparticles treatment (2−25 μg/mL) significantly increases the proliferation of rat adipose stem cells and bone marrow stem cells [85]. The induction of proliferation and osteogenic differentiation of mesenchymal stem cells may interfere with metastasis of osteosarcoma. However, further study is required to confirm the role of mesenchymal stem cells and their therapeutic potentials in osteosarcoma.

Regarding anticancer selectivity, selenium selectively exerts anticancer actions on osteosarcoma cells. Both inorganic and organic selenium exhibit no or marginal toxicity on several primary and non-tumourigenic cells including primary rat growth plate chondrocytes [86], primary mouse lung fibroblasts [57], primary human calvarial osteoblasts [87], human bone marrow stem cells (BMSCs) [60,88], human umbilical cord stem cells [84], human lymphocytes [61], mouse preosteoblast MC3T3-E1 cells [57,58,86], mouse 3T3-L1 preadipocytes [89], rat skeletal muscle L6 cells [50], human embryonic kidney 293 cells [50] and human osteoblast hFOB1.19 cells [56,59]. Additionally a low concentration (0.01–1 µM) and 1 h transient treatment of SeO_3_^2−^ exert radioprotective effects on chondrocytes and osteoblasts by protecting them from 20-Gy irradiation-induced cytotoxicity [86]. SeO_3_^2−^ (0.01–1 µM) is also not genotoxic to human blood lymphocytes and does not cause mitotic index change, chromosomal aberration or chromatid break [61]. However, SeO_3_^2−^ (1 µM for 1 h) substantially increases the frequency of dicentric chromosomes (but not deletion and chromatid break) in γ rays-irradiated human blood lymphocytes via an unknown mechanism [61]. Additionally, selenium nanoparticles (500–6000 nm size, 0.005 mg/mL for 1 week) also induce significant cytotoxic effects and apoptosis events on mesenchymal stem cells via morphological observation [90]. These findings require further investigation to ensure the safety of selenium in daily consumption.

The molecular mechanisms on the selectivity of selenium have not been studied in detail in osteosarcoma cell models. Generally, cancerous cells are more sensitive to oxidative stress compared with normal cells [91,92]. This characteristic has been exploited by current chemotherapeutic agent, such as doxorubicin, to selectively induce osteosarcoma cell death via oxidative stress [93,94]. Therefore, the selective cytotoxicity of selenium toward osteosarcoma cells may be due to oxidative stress. A high concentration of inorganic or organic selenium induced non-selective cell death on non-tumourigenic cells and osteosarcoma cells [50,52,57,62], probably due to the overwhelming ROS production.

### 3.4. Selenium as a Bone Implant Material

Selenium serves as a potential candidate for bone implant material, especially for patients with osteosarcoma. In vitro osteogenic studies were conducted on selenium-doped titanium substrates [56,87], calcium phosphate [58], poly-L-lactic acid (PLLA) nanocomposites [59] or HA [57,60,62,66]. These selenium (SeO_3_^2−^)-doped substrates suppress the growth of mouse and human osteosarcoma cells including K7M2 [56,57], MG-63 [58,59,60], Saos-2 [62] and MNNG/HOS cells [66]. In parallel with the in vitro studies, selenium-substituted HA substrate and nanoparticles induce apoptosis of osteosarcoma cells with early ROS generation, mitochondria dysfunction, cytochrome *c* release, tBid upregulation and caspase-8,-9,-3 activation [60,66]. The addition of manganese (III) tetrakis(1-methyl-4-pyridyl)porphyrin (MnTMPyP; an oxidative stress inhibitor) inhibits selenium-substituted HA nanoparticles-induced ROS generation and osteosarcoma apoptosis [66], thus, further emphasising the importance of ROS in anticancer activities of selenium. The addition of SeO_3_^2−^ into bone implant substrates does not affect their basic reaction chemistry [56]. Physicochemical analysis revealed that SeO_3_^2−^ is dose-dependently released from the substrate surface, thereby explaining its anticancer effects [56]. Similarly, acellular media collected from SeO_3_^2−^-doped substrate are proven effective against mouse osteosarcoma cells [56]. Wang et al. reported that SeO_3_^2−^-doped HA nanoparticles are directly internalised into the osteosarcoma cells via nonspecific endocytosis [66]. SeO_3_^2−^ in the vesicles is then released into the cytosol upon the degradation of HA nanoparticles during the merger of endosome and acidic lysosome in a pH-dependent manner [66]. The release of SeO_3_^2−^ increases ROS generation, leading to osteosarcoma cell apoptosis in 6% and 10% SeHANs and 2µM SeO_3_^2−^, but not in HA nanoparticles control and 3% SeHAN groups [66].

SeO_3_^2−^-doped substrates also showed osteoinductive activities by promoting the growth of non-cancerous human osteoblast hFOB 1.19 cells [56], non-cancerous human preosteoblast MC3T3-E1 cells [58] and primary human calvarial osteoblasts [87]. SeO_3_^2−^-doped substrates increased the growth of mouse primary lung fibroblasts [57] and BMSCs [60] under a similar treatment. SeO_3_^2−^-doped titanium substrate [56], selenium (SeO_3_^2−^)-doped calcium phosphate coating [58] and SeO_3_^2−^coated-PLLA nanocomposites (SeNP-PLLA) [59] promote bone-forming activities, as evidenced by the increase in alkaline phosphatase activities and extracellular calcium deposition. Furthermore, SeO_3_^2−^-substituted HA exhibits osteoinductive activity on partially differentiated MC3T3-E1 cells by increasing the mRNA expression of bone γ-carboxyglutamate protein 3 (BGLAP3; osteocalcin-related protein) [57].

The cytotoxicity of SeO_3_^2−^-doped substrates and nanoparticles relies on its pH-dependent release into the medium and a high concentration of selenium is suggested to induce non-selective cytotoxicity [57,59,62]. A high SeO_3_^2−^ content (3.0 wt%) induces non-cancerous MC3T3-E1 cell death with abnormal morphological changes as early as 24 h of treatment [57]. SeO_3_^2−^-containing HA (SeHA) 3.0 wt% treatment also decreases BGLAP3 expression due to its cytotoxic effect [57]. SeNP-PLLA reduces the viability of human osteosarcoma MG-63 cells and non-cancerous hFOB cells, though the effects are more selective on former than on the latter [59]. Additionally, the media after overnight incubation with selenium-containing hydroxyapatite/alginate (SeHA/ALG) composite microgranules are cytotoxic to human osteosarcoma Saos-2 cells and non-cancerous hFOB 1.19 cells with almost 90% reduction on viability [62]. According to the authors, this non-selective cytotoxicity of SeHA/ALG microgranules may be due to the rapid release and accumulation of selenium in the culture media [62]. As previously discussed, excessive ROS production is one of the mechanisms of selenium in inducing unspecific cell death [95,96].

The in vitro studies of selenium and its derivatives on osteosarcoma cells are summarised in Table 1.

## 4. Evidence from Animal and Human Studies

Several animal studies were conducted to determine the effects of selenium on osteosarcoma [49,51,55,66,97,98]. Bierke and Svedenstal initiated the study on the effects of inorganic selenium in radioactive strontium (^90^Sr)-induced osteosarcoma mice [98]. Vitamin E (α-tocopherol acetate) with or without SeO_3_^2−^ (10 µg) was administrated intraperitoneally to the mice with osteosarcoma every 2 weeks from day 105 after the ^90^Sr exposure until 14 months [98]. The same injection was continued after 14 months but was changed to 30-day intervals for the rest of the life span [98]. Oestrogen (polyestradiol phosphate) was administered in certain groups during 30, 60 and 90 days after the ^90^Sr exposure. The results showed no significant difference in osteosarcoma tumour incidence after treatment with vitamin E with or without SeO_3_^2−^. Post-exposure of antioxidants, including selenium and vitamin E, is not beneficial to prevent the development of ^90^Sr-induced osteosarcoma. The combined treatment somehow hastened the onset of osteosarcoma regardless of oestrogen induction [98].

Several studies of inorganic and organic selenium were conducted using osteosarcoma xenograft animal models [49,51,55,66,97]. Some in vitro human osteosarcoma cell lines were used, including human osteosarcoma KOS [49], U-2 OS [51], MG-63 [55] and SOSP-9607 cells [97]. Hiraoka et al. investigated the effect of SeO_2_ in BALB/c nude mice implanted with osteosarcoma xenograft [49]. The back of nude mice was subcutaneously inoculated with KOS cells, and the mice were fed with SeO_2−_containing drinking water (0.2 and 2 µg/mL) until day 44 after inoculation [49]. SeO_2_ dose-dependently decreased the tumour volume to 2.5-fold lower compared with that of control [49]. SeO_2_ also induces apoptosis in xenograft tumour tissues without affecting visceral organs. However, this compound does not prevent tumour incidence [49].

For the organic selenium, Wang et al. reported that daily oral administration of Se-Poly isolated from Ziyang green tea (200 and 400 mg/kg) for 28 days significantly reduces tumour volume and weight of U-2 OS xenograft in BALB/c nude mice [51]. Similar to that used by Hiraoka et al., the Se-Poly is non-toxic to nude mice where it does not affect the body weight or cause any lethal incidence [51]. Wang et al. showed that the intravenous injection of SeC (5 and 10 mg/kg; every other day for 2 weeks) significantly and dose-dependently reduces the osteosarcoma MG-63 tumour xenograft volume and weight in nude mice [55]. Mechanistically, SeC induces p53 phosphorylation (Ser 15) and caspase-3 activation in tumour xenograft in a dose-dependent manner [55]. SeC also significantly suppresses cell proliferation and angiogenesis of tumour xenografts as evidenced by the downregulation of Ki-67 and CD-34 biomarkers [55]. Similar to other selenium species, SeC does not affect the body weight, suggesting the lack of systemic toxicity in nude mice [55].

The anticancer effect of SeO_3_^2−^-doped substrates on osteosarcoma xenograft animal models has also been reported [66,97]. Wang et al. revealed that intratumoural injection of SeHAN for 30 days significantly reduces the tumour volume of intrafemoral human osteosarcoma SOSP-9607 xenograft in nude mice [97]. SeHAN also inhibits osteosarcoma tumour metastasis into the lung and protects other vital organs, such as liver, kidney and cardiac muscles from osteosarcoma-mediated damages [97]. The anticancer effect of SeHAN is mediated by the suppression of tumour invasion but not proliferation, as indicated by the reduction of matrix metallopeptidase-9 (MMP-9; invasion marker) and the lack of change in Ki-67 level (mitotic marker) [97]. Intratumour 10% SeHAN injection (every 3 days for 30 days) significantly reduces the tumour size, weight and volume of osteosarcoma MNNG/HOS tumour xenograft in BALB/c nude mice [66]. SeHAN induces oxidative DNA damage, which will hypothetically trigger the subsequent activation of caspases and the apoptosis in tumour tissues [66]. In parallel with in vitro studies, the anticancer effect of SeHAN is related to the release of SeO_3_^2−^ ions into aqueous solution [97]. SeHAN is completely degraded within tumour tissues with less calcium aggregation and blood vessel vascularization upon histological analysis [66]. Similar to findings in other selenium studies, SeHAN does not cause any significant systemic toxicity in nude mice and has no effect on body weight, lethality, haematological indices and serum biochemical profile, including aspartate aminotransferase, blood urea nitrogen, creatinine and lactate dehydrogenase levels [66,97]. Furthermore, no pathological change has been detected in the liver of nude mice that received SeHAN treatment [66].

One human study was conducted by Huang et al. to identify the relationship between selenium level and osteosarcoma disease [54]. No significant difference was found in the plasma selenium levels between patients with and without osteosarcoma [54]. Selenium levels were significantly higher in osteosarcoma tissues compared with those in normal bone tissues among patients with osteosarcoma [54]. However, further investigation is needed to identify the role of high selenium levels in osteosarcoma tissues. To date, no human study has revealed the beneficial effect of selenium supplementation in preventing osteosarcoma and no clinical trial is being conducted to evaluate the therapeutic effect of selenium in patients with osteosarcoma.

Several epidemiological studies and clinical trials were conducted to determine the relationship between selenium intake and the risk of other solid cancers; however, the findings are heterogeneous [5,34,99,100,101,102,103,104,105]. The NPC and Linxian Nutritional Intervention Trials reported that selenium intake reduces the risk of lung, colorectal and prostate cancer and mortality related to oesophageal and gastric cancer [33,34,35,36]. The Selenium and Vitamin E Cancer Prevention Trial (SELECT) and Selenium and Celecoxib (Sel/Cel) Trial showed that selenium does not reduce the risk of prostate [33,101] and colorectal cancer [106]. Additionally, a recent systematic review and meta-analysis by Vinceti et al. concluded that selenium supplementation does not reduce the overall cancer incidence or mortality [103,107]. The contradicting findings may be confounded by experimental biases, chemical forms of selenium, basal selenium status, nutritional status and lifestyle factors of the subjects [35,107]. Vinceti et al. emphasised on randomised controlled clinical trials of selenium on various cancers [103]. The osteosarcoma is not involved as currently the relevant clinical trial is not available. Further studies are required to confirm the relationship of selenium and cancer risk, especially on osteosarcoma. Table 2 summarises the effects of selenium in osteosarcoma in vivo.

## 5. Conclusions

Selenium and selenium-containing proteins possess potential anticancer activity, as evidenced from cellular and animal studies. Evidence was provided that the underlying mechanism for their anticancer effects involves increased intracellular ROS generation, leading to activation of the p53/ATM/FOXO3a pathway, proapoptotic proteins and caspases. This results in cytotoxicity, antiproliferative and proapoptotic effects of selenium compounds on osteosarcoma. The action of selenium can be selective on osteosarcoma cells without affecting adjacent normal cells, such as chondrocytes and fibroblasts. Moreover, this element is osteogenic for normal bone tissues. Developing SeO_3_^2−^-doped bone biomaterials, which release this essential element in a controlled manner for therapeutic purposes, remains a challenge because SeO_3_^2−^ at a high concentration exerts non-specific cell death via oxidative stress. Another challenge is identifying the most active selenium form and dose that exert the best anti-osteosarcoma effects in the various cellular models used in previous experiments. In vivo studies showed that selenium can reduce the tumour size in animals with osteosarcoma, but does not diminish the incidence of the tumour. However, the effects of selenium on osteosarcoma have not been validated in a clinical trial. A paucity of epidemiological data revealed a relationship between selenium intake and osteosarcoma. Thus, the role of inorganic and/or organic selenium on osteosarcoma tumour formation remains unknown. These gaps in our knowledge should be filled by researchers to determine the role of selenium in preventing or treating osteosarcoma.

## Figures and Tables

**Figure 1 ijms-20-05318-f001:**
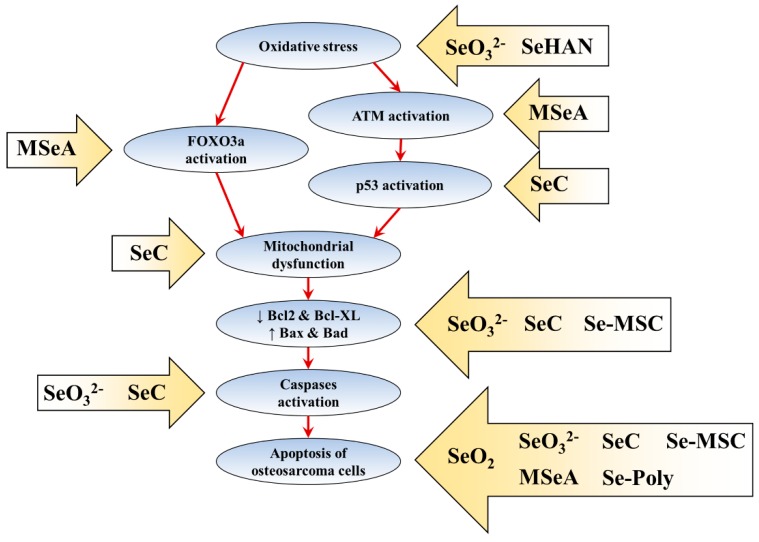
The molecular mechanisms of selenium-induced osteosarcoma cell death. The arrow boxes with selenium species showed the reported effects of selenium species on the apoptosis pathway. The red arrows indicate the sequent of events in apoptosis pathway.

**Table 1 ijms-20-05318-t001:** The anticancer effects of selenium from in vitro studies.

Selenium Types	In vitro Models	Treatment Condition	Results	Ref
SeO_2_	Human osteosarcoma MG-63, U-2 OS and Saos-2 cells	0.02−20 µg/mL (equivalent to 0.18–180.25 µM) for 24 h	Inhibited osteosarcoma cells proliferation in concentration (18.02 µM and above) and time-dependent (4 h and above) manner Induced apoptosis with morphological and ultrastructural changes	[49]
SeO_3_^2−^	Primary rat growth plate chondrocytes and non-tumourigenic mouse osteoblast MC3T3 cells	Treatment with SeO_3_^2−^ (0.005–25 µM). Treatment time not indicated.	Non-cytotoxic up to 25 µM	[86]
		Pretreatment with 0.05 µM SeO_3_^2−^. Treatment time not indicated.	Protected primary rat growth plate chondrocytes and mouse osteoblast MC3T3 cells from 20-Gy irradiation-induced cytotoxicity	
	Human osteosarcoma U-2 OS cells	Pretreatment of SeO_3_^2−^ (0.01–10 µM) for 1 h, followed by 24 h cisplatin treatment	SeO_3_^2−^ was non-cytotoxic to U-2 OS cells up to 10 µM Protected U-2 OS cells from cisplatin-induced cytotoxicity Reduced the transcription-coupled repair pathway	[61]
	Human blood lymphocytes	Pretreatment of SeO_3_^2−^ (0.01–10 µM) for 1 h, followed by 150 cGy γ irradiation	Cytotoxic to lymphocytes at the highest concentration (10 µM) Did not exert genotoxic effect or alter mitotic index up to 1 µM	
	Human osteosarcoma U-2 OS cells, non-tumourigenic human embryonic kidney 293 cells and rat skeletal muscle L6 cells	Treated (5–40 µM) for 24–48 h	Induced early ROS production (1 h) in U-2 OS cells via a concentration and time-dependent manner Increased proapoptotic genes mRNA levels (*PTEN*, *P53*, *BAX*, *CASP9* and *CASP3*) and decreased antiapoptotic gene *BCL-2* mRNA level in U-2 OS cells (20 µM) after 48 h treatment Increased proapoptotic proteins (Bax and caspase-3) expression and reduced anti-apoptotic Bcl-2 protein level in U-2 OS cells Induced U-2 OS cell death (IC_50_ of 20 µM) with typical apoptotic morphology changes after 24-h treatment Suppressed the 12−days colony formation of U-2 OS cells at 10 µM No/less cytotoxic to 293 and L6 cells in similar concentrations	[50]
Se-Poly	Human osteosarcoma U-2 OS cells	Treated with 25–200 µg/mL of purified Se-Poly for 24 to 72 h	Treatment ≥ 50 µg/mL was cytotoxic to U-2 OS cells in a concentration-dependent manner, whereby LDH leakage was detected 24 h post treatment	[51]
MSeA	Human osteosarcoma U-2 OS cells	Treated with MSeA (0–10 µM) for 48 h with or without the knockdown of WRN	Induced cytotoxicity (IC_50_ value of 4 µM) after 48-h treatment Significantly increased the event of necrosis at 4 µM concentration after 48-h treatment Increased the phosphorylation of ATM but not H2Ax ATM inhibitor (KU55933) reduced the MSeA-induced cytotoxicity Knockdown of WRN further potentiated MSeA-induced cytotoxicity	[52]
		Treated with 1.5 and 5 µM MSeA up to 72 h	Significantly inhibited the growth of U-2 OS cells that stably expressed with FOXO3a protein Induced nuclear translocation of FOXO3a protein after 6-h treatment Knockdown of FOXO3a suppressed the MSeA-induced ROS reduction, G1 arrest, cytotoxicity and apoptosis	[53]
Se-MSC	Human osteosarcoma MG-63 and U-2 OS cellsDrug-resistant human osteosarcoma Saos-2/MTX300 cells	Treated with Se-MSC (0–150 µM) up to 96 h	Slightly promoted the growth of U-2 OS cells Induced cytotoxicity, G1 arrest and apoptosis in MG-63 cells Induced downregulation of Bcl-2 protein and upregulation of Bax protein in MG-63 cells	[54]
		Treated with Se-MSC (0–250 µM) up to 96 h	Inhibited the growth of Saos-2/MTX300 cells in a concentration and time-dependent manner	
SeC	Human osteosarcoma MG-63 cells	Treated with SeC (0–20 µM) up to 72 h	Cytotoxic to MG-63 cells in a concentration and time-dependent manner Induced S-phase arrest and downregulated of cyclinA and CDK-2 proteins Induced MG-63 cells apoptosis with early mitochondrial membrane potential loss, mitochondrial fragmentation, ROS production, p53 activation, upregulation of Bax and Bad proapoptotic protein, downregulation of Bcl-2 and Bcl-XL antiapoptotic protein, caspase-9 activation, caspase-3/7 activation and PARP cleavage SeC-mediated ROS production in p53 activation	[55]
SeO_3_^2−^-doping titanium substrate	Mouse osteosarcoma K7M2−pCl Neo cells and non-tumourigenic human osteoblast hFOB 1.19 cells	Cultured on selenite-doping titanium surface for 4, 24 and/or 72 h	Reduced mouse osteosarcoma cell density after 4- and 72−h incubation Increased the normal osteoblast cell density, ALP activity and extracellular calcium deposition of hFOB 1.19 cells after 4- and 24-h treatment	[56]
		Cultured K7M2−pCl Neo cells and hFOB 1.19 cells with acellular media that collected from selenium-doping orthopaedic implant material (3 days incubation) for 3 days	Selenium was dose-dependently released into culture media after 2−3 days incubation Significantly decreased the cellular density of mouse osteosarcoma cells but had no effect on normal osteoblast cell density	
	Mouse osteosarcoma osteoblast from ATCC (unknown cell types) and normal primary human calvarial osteoblast	Cultured on SeO_3_^2−^-doping titanium surface for 3 days	Decreased the mouse osteosarcoma osteoblast density and promoted the growth of normal osteoblast	[87]
		Both cancerous and normal osteoblast were co-cultured on the uncoated or SeO_3_^2−^-doping titanium surfaces for 4–65 h	Suppressed the growth of osteosarcoma cells and increased the normal osteoblast density after 53 and 65 h of co-culturing	
Selenium-doped calcium phosphate coating	Human osteosarcoma MG-63 cells and non-tumourigenic preosteoblast MC3T3-E1 cells	Cultured the cells with selenium-doped coatings (0.6 and 2.7 at% selenium) for 24 h or up to 21 days incubation	All coatings are non-cytotoxic to MC3T3-E1 cells up to 21 days incubation 2.7at% selenium-doped coating induced osteogenic activity by increasing ALP activity in MC3T3-E1 cells after 21 days incubation 2.7at% selenium-doped coating suppressed the growth of osteosarcoma cells and promoted the growth of MC3T3-E1 cells as early as 24 h of treatment. Selenium-doped coating with 0.6at% selenium gave a similar finding only after 72 h of treatment	[58]
SeNP-PLLA nanocomposites	Human osteosarcoma MG-63 cells and non-tumourigenic foetal osteoblast hFOB cells	Incubated with SeNP-PLLA (0.025M SeO_3_^2−^) for 48 h	Increased normal osteoblast bone forming activity marked by increased ALP activities Reduced both osteosarcoma cells and normal osteoblasts but more selective on osteosarcoma cells with greater reduction of growth rate	[59]
SeHAN	Human osteosarcoma MG-63 cells and normal human BMSCs	SeHAN with different preparation concentration and treatment concentration (50–200 µg/mL) was added into culture medium with a monolayer of osteosarcoma cells for 3 days	SeHAN prepared from 0.882mM SeO_3_^2−^ (concentration not indicated) induced osteosarcoma cell death and supported the growth of normal BMSCs after 48−, 60− and 72−h incubation Induced cytotoxicity (100–200 µg/mL) and apoptosis (200 µg/mL) in MG-63 cells after 48-h incubation Induced early ROS production (30 min) and mitochondrial transmembrane potential loss (48 h) on MG-63 cells after 200 µg/mL treatment	[60]
	Human osteosarcoma MNNG/HOS cells	SeHANs (3, 6 and 10% molar ratio of selenium and phosphate) in pH 5.0 and pH 7.4 were added to a monolayer of osteosarcoma cells at 50 µg/mL up to 24 h (2 µM sodium SeO_3_^2−^ as control)	More selenium was released from SeHANs in the acidic cell-free solution than neutral solution Uptake of SeHANs by osteosarcoma transpired through endocytosis. Release of the intracellular selenium occurred via pH-dependent degradation of SeHANs during the merging of endosome and acidic lysosome SeHAN (6 and 10%) and SeO_3_^2−^ increased early ROS production (6–24 h) and led to MNNG/HOS cell apoptosis after 48 h treatment MnTMPyP inhibited SeHAN (6 and 10%) and SeO_3_^2−^-induced ROS production and apoptosis in MNNG/HOS cells 10% SeHAN induced the release of mitochondrial cytochrome *c*, caspase-8 activation, tBid upregulation, caspase-9 activation and caspase-3 activation as early as 12 h treatment	[66]
SeHA	Mouse osteosarcoma K7M2 cells and normal primary mouse lung fibroblast (isolated from C57B/6/J mouse lungs)	SeHA (0.102−3.0 wt%) in 2 mg/mL were added to monolayer cells for 48 h	SeHA (1.922 and 3.0 wt%) significantly reduced the viability of osteosarcoma cells and promoted the growth of normal fibroblasts	[57]
	Non-tumourigenic mouse calvarial preosteoblastic MC3T3-E1 subclone 4 cells	SeHA (0.102−3.0 wt%) in 5 mg/mL were added to fibroblast for 72 h	Exerted osteoinductive activity marked by increased BGLAP3 mRNA level (1.238–3.0 wt% SeHA) Increased GAPDH mRNA level with the highest induction at 1.922 wt% SeHA High SeO_3_^2−^ content (3.0 wt%) was cytotoxic to MC3T3-E1 cells	
SeHA/ALG	Human osteosarcoma Saos-2 cells and non-tumourigenic hFOB 1.19 cells	Treated with supernatant extracts (50 and 100%) from SeHA/ALG, SeHA/ALG/RIS I and II for 24 h	Non-selective cytotoxicity (50 and 100% supernatant extracts) to both osteosarcoma cells and normal osteoblasts, marked by > 90% reduction in viability	[62]

**Table 2 ijms-20-05318-t002:** The effects of selenium in osteosarcoma animal and human studies.

Selenium Types	In vivo Models and Treatment	Positive Results	Negative Results	Ref
SeO_2_	BALB/c nu/nu mice with subcutaneous human osteosarcoma KOS xenograft were fed with SeO_2−_containing drinking water (0.2 and 2 µg/mL) until day 44 after inoculation	Tumour xenograft volume was reduced (2 µg/mL treatment) via apoptosis induction Did not induce apoptosis on other visceral organs	Did not suppress the tumour incidence	[49]
SeO_3_^2−^	α-tocopherol acetate (0.5 mg) with or without SeO_3_^2−^ (10 µg) was injected intraperitoneally into ^90^Sr-induced osteosarcoma mice every 2 weeks, starting from day 105 after exposure until 14-month. The regime changed to 30-day intervals for the rest of the life-span.		Post-exposure of antioxidants like α-tocopherol acetate and SeO_3_^2−^ did not prevent the development of ^90^Sr-induced osteosarcoma.	[98]
Se-Poly	U-2 OS xenograft (100 mm^3^) BALB/c nude mice were orally treated with Se-Poly from Ziyang green tea (100, 200 and 400 mg/kg) daily for 28 days.	Tumour volume and tumour weight were reduced at 200 and 400 mg/kg treatment without affecting body weights or cause any lethality		[51]
SeC	Nude mice with MG-63 xenograft (50 mm^3^) was treated with intravenous injection of SeC (5 and 10 mg/kg/day) every other day for 2 weeks	Tumour xenograft volume and weight were reduced Tumour xenografts cell proliferation and angiogenesis were suppressed p53 phosphorylation and caspase-3 activation were induced Body weight of nude mice was reduced		[55]
SeHAN	Nude mice with an orthotopic intrafemorally injection of SOSP-9607 xenograft were treated with intratumoural injection of SeHAN for 30 days	Tumour volume of xenograft was reduced Tumour metastasis into the lung was inhibited Tumour invasion but not proliferation was suppressed Other organs (liver, kidney and cardiac muscles) were protected from osteosarcoma-mediated damages Good compatibility with no or lesser effect on lethality, systemic toxicity, haematological indices and biochemical profile		[97]
	BALB/c nude mice with osteosarcoma xenograft (100 mm^3^) were treated with intratumoural injection of 10%SeHAN every 3 days for 30 days	Tumour size, weight and volume of xenograft were reduced via apoptosis induction with oxidative DNA damage and caspases activation No effects on lethality, body weight, pathological liver changes and serum biochemical profile SeHAN was completely degraded within tumour tissues with lesser calcium aggregation and vascularization		[66]
Selenium levels	Nine osteosarcoma patients and nine non-osteosarcoma patients		No significant difference in the serum selenium levels between osteosarcoma and non-osteosarcoma patients	[54]
	Paired osteosarcoma and normal bone tissues from 14 osteosarcoma patients		Higher selenium levels in osteosarcoma bone tissues	

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
