# Peer review of "Emerging Anticancer Potentials of Selenium on Osteosarcoma"

_ijms, 2019, doi:10.3390/ijms20215318_

Round 1

Reviewer 1 Report

The review entitled "Emerging Anticancer Potentials of Selenium on Osteosarcoma" reports pieces of evidence about the anticancer activity of inorganic and organic selenium on osteosarcoma. The topic is interesting and clearly written, however, the authors did not discuss the role of inorganic and organic selenium on tumour formation. In particular, could be interesting to provide an overview on selenium activity on the different cells of the tumour microenvironment that influence osteosarcoma progression, and, in particular, on osteosarcoma progenitor cells, i.e. mesenchymal stem cells. This is an important issue that would enhance the impact of the review. 

Author Response

Thank you for your constructive comments.

The review entitled "Emerging Anticancer Potentials of Selenium on Osteosarcoma" reports pieces of evidence about the anticancer activity of inorganic and organic selenium on osteosarcoma. The topic is interesting and clearly written, however, the authors did not discuss the role of inorganic and organic selenium on tumour formation.

Reply: We thank reviewer for the comments. Currently information on the role of inorganic or organic selenium on the formation of osteosarcoma is lacking. There is no direct comparison between inorganic and organic selenium in in-vitro and in-vivo models. In term of osteosarcoma tumour formation, there is only one relevant study by Hiraoka et al. 2001, demonstrating that inorganic SeO2-containing drinking water reduced osteosarcoma tumor volume but not the tumor incidence (line 305-311). Additionally, no human study has been conducted to identify the role of selenium supplementation (regardless inorganic or organic selenium) in preventing osteosarcoma formation. We had highlighted this research gap in the conclusion (line 382-383).

In particular, could be interesting to provide an overview on selenium activity on the different cells of the tumour microenvironment that influence osteosarcoma progression, and, in particular, on osteosarcoma progenitor cells, i.e. mesenchymal stem cells. This is an important issue that would enhance the impact of the review.

Reply: We thank reviewer for the suggestions. We discussed the role of osteoclasts and mesenchymal stem cells in osteosarcoma. We also included the effects of selenium on osteoclast and mesenchymal stem cells (line 163-194).

Reviewer 2 Report

The manuscript by Kok-Lun Pang et al is a review about the anticancer effect of Selenium on osteosarcoma. The authors report experimental studies aimed to identify the effects of selenium on the survival of osteosarcoma cells, both in vitro and in vivo experimental models, and human patients.

The paper looks to cover the main body of the related literature.

Some aspects should be addressed to endorse the scientific soundness of the paper:

1) RANKL pathway is important in the progression and dissemination of osteosarcoma. Authors should report and discuss whether selenium affects somehow rankl/opg/rank axis.
2) In lines 68- 71 osteosarcoma incidence is reported. A better description should be addressed, highlighting more that osteosarcoma is a rare bone tumour.
3) The authors report a review article on osteosarcoma/selenium from Vinceti et al, 2018. Since the topic is very narrow, the authors must describe the main differences between the two papers and if the same original studies are commented, highlighting similarities/divergences and updates.

Author Response

Thank you for your constructive comments. Please kindly find our response in the attached response sheet. 

Round 2

Reviewer 1 Report

The Authors answered all my questions. The manuscript is now suitable for publication.